# Synthesis and Biological Evaluation of Fangchinoline Derivatives as Anti-Inflammatory Agents through Inactivation of Inflammasome

**DOI:** 10.3390/molecules24061154

**Published:** 2019-03-23

**Authors:** Ting Liu, Qingxuan Zeng, Xiaoqiang Zhao, Wei Wei, Yinghong Li, Hongbin Deng, Danqing Song

**Affiliations:** 1Institute of Medicinal Biotechnology, Chinese Academy of Medical Sciences and Peking Union Medical College, Beijing 100050, China; lutyliu@126.com (T.L.); zqx50810793@163.com (Q.Z.); xiaoqiangzhao2017@126.com (X.Z.); weiwei082695@163.com (W.W.); songdanqingsdq@hotmail.com (D.S.); 2Institute for Food and Cosmetics Control, National Institutes for Food and Drug Control, Beijing 100050, China

**Keywords:** fangchinoline, Anti-inflammatory, IL-1β, NLRP3 inflammasome

## Abstract

Twenty eight 7-substitued fangchinoline analogues, of which twenty two were novel, were synthesized and evaluated for their effect to inhibit lipopolysaccharide/nigericin (LPS/NIG)-induced IL-1β release at both cell and protein levels at the concentration of 5 μM. Among them, compound **6** exhibited promising inhibitory potency against IL-β activation with an IC_50_ value of 3.7 μM. Preliminary mechanism study revealed that **6** might target NLRP3 protein, and then block ASC pyroptosome formation with-NLRP3, rather than acting on the activation of the NLRP3 inflammasome (NF-κB and MAPK pathways) or caspase-1 protein. Our current study supported the potential role of compound **6** against IL-β activation, and provided powerful information for developing fangchinoline derivatives into a novel class of anti-inflammatory agents.

## 1. Introduction

Inflammation is an indispensable immunological response to tissue injury induced by a wide variety of stimuli, such as pathogens, damaged cells, or irritants [1,2,3], and is a protective response involving immune cells, blood vessels, and molecular mediators [4]. However, inappropriate inflammation generates excess amounts of inflammatory cytokines which would cause tissue destruction and ultimately result in severe disorders, such as gout, atherosclerosis, type 2 diabetes, metabolic syndrome, age-related macular degeneration, Alzheimer’s disease, multiple sclerosis, and inflammatory bowel disease [2,3]. Unfortunately, for most of these conditions, no satisfactory treatment is available.

As the most characterized pro-inflammatory cytokine among IL-1 family members, IL-1β is a potent pro-inflammatory mediator in many immune reactions, and IL-1β antibody has demonstrated great promises in the treatment of inflammatory-related disorders including cardiovascular and rheumatologic [5,6,7,8]. Abundant studies have indicated that the nucleotide-binding domain (NOD)-like receptor protein 3 (NLRP3) inflammasome, which consists of the sensor protein NLRP3 (also known as NALP3), the pro-inflammatory mediator caspase-1, and the adaptor protein apoptosis-associated speck-like protein (ASC), is responsible for the production of IL-1β [9,10]. The activation of NLRP3 inflammasome requires dual signals [11]. One, referred to as priming, is the nuclear factor kappa-light-chain-enhancer of activated B cells (NF-κB)-dependent production of pro-IL-1β and NLRP3 [12,13]. The other is the narrower activation of NLRP3 inflammasome [11], in which-NLRP3 recruites the oligomerized ASC mediated by mitogen-activated protein kinase (MAPK) pathway [14], and pro-caspase-1, which is responsible for the cleavage of pro-IL-1β into IL-1β. Therefore, the inhibition of NLRP3 mediated IL-1β secretion is now generally considered as an effective strategy to prevent inflammation associated diseases [10,15].

In recent years, our group has been dedicated to the exploration and discovery of new anti-inflammatory agents from a natural alkaloid library [16]. Since THP-1 cell line is readily to form NLRP3 inflammasome under the stimuli of lipopolysaccharide (LPS) and nigericin (NIG) [17], we established a NLRP3-mediated IL-1β activation screening model in THP-1 cells. Then, a Chinese natural bisbenzylisoquinoline alkaloid, fangchinoline (**1**, Figure 1) was found by screening to exert a moderate anti-inflammatory effect with the inhibitory rate of 50.5% (5 μM). It was reported that **1** could reduce the incidence and severity of LPS-induced inflammation reactions *in vivo* with an unknown mode of action [18]. The unique bisbenzylisoquinoline skeleton of **1** prompted us to investigate the anti-inflammatory effect of this compound class, and 7*O*-methylfangchinoline (**2**) named tetrandrine (Figure 1), was found to retain the activity by inhibiting IL-1β release at a rate of 40.7% (5 μM), indicating the potential promise of its kind as anti-inflammatory agents. The free hydroxyl group in compound **1** provides a ready handle group for subsequent structural modifications. Therefore, in the present study, as illustrated in Figure 1, taking **1** as the lead, a series of bisbenzylisoquinoline analogues with diverse substitution patterns at the C7 position were designed, synthesized and evaluated for their inhibition of NLRP3 mediated IL-1β activation triggered by LPS/NIG in THP-1 cells. The anti-inflammatory mechanism study of key compounds was carried out as well.

## 2. Results and Discussion

### 2.1. Chemistry

A total of 28 bisbenzylisoquinoline derivatives were prepared using commercially available fangchinoline (**2**) with purity over 95% as the starting material, which was purchased from Xi’an Tianbao Biotechnology Co., Ltd. (Shanxi, China). Compounds **3a**–**c** [19,20], **3e**–**g** [19,21], **4a** [20] and **4b** [20] were prepared according to literature methods. As shown in Scheme 1, 7*O*-alkyl fangchinoline analogues **3d** and **3h** were obtained in reasonable yields via the reactions of **2** and various alkyl or aromatic halides in DMF, using sodium hydride as the acid neutralizer [19,20,21]. 7*O*-Acyl fangchinoline analogues **4c**–**e** were obtained in 35–52% yields by acylation of **1** with various acyl chloridew, using triethylamine as the base [20]. 7*O*-Sulfonylfangchinoline products **5a** and **5b** were obtained in 33–50% yields by sulfonation of **1** in a similar method [20].

The reaction of **2** and *N*-(3-bromopropyl)phthalimide in DMF and sodium hydride under an inert atmosphere provided phthalimide **6** in 50% yield, which was converted in a high yield into its primary amine derivative **7** with hydrazine via a Gabriel reaction [22]. The acylation or sulfonation of **7** gave the corresponding target compounds **8a**–**j** or **8k**–**m** in 22–71% yields. All the final products were purified by flash column chromatography on silica gel using CH_2_Cl_2_/CH_3_OH as the eluent.

### 2.2. Biological Activity

All analogues were examined in THP-1 cells for their inhibitory effects on LPS/NIG-triggered IL-1β activation at the concentration of 5 μM by ELISA assay. Structures of the analogues and their inhibitory rates on IL-1β release activity were shown in Table 1. Our SAR analysis was focused on the influence of the substituents on the 7-position of fangchinoline. As a start, the activity of nine ether derivatives **2** and **3a**–**h** was tested and compared. Apparently, the length of the ether motif affected the activity. The methyl (**2**), ethyl (**3a**), isobutyl (**3c**) and ethyloxyethyl (**3d**) ethers gave slight decreases in activity, while the propyl ether **3b** exhibited a significantly improved inhibitory rate of 71.8%. The introduction of a cyclic substituent was beneficial for activity, for example, saturated cyclopropyl methyl and cyclohexyl methyl ethers **3e** and **3f** showed improved inhibitory rates of 67.4% and 66.0%, and unsaturated benzyl and pyridylmethyl ethers **3g** and **3h** also gave higher inhibitory rates of 80.0% and 69.3%, respectively.

Then a series of carbonyl ester derivatives **4a**–**e** and two sulfonyl ester derivatives **5a** and **5b** were also evaluated, and all of them displayed comparable or improved activities. Among them, 3,3-dimethylbutyryl and *m*-chlorobenzenesufonyl derivatives **4e** and **5b** gave the top inhibition rates of 72.1% and 80.5%, respectively.

Considering that compound **3b** bearing a propyl ether gave the highest inhibition rate among the ether derivatives, novel aminopropyl ethers, including 7-phthalimidepropylfangchinoline (**6**) and 7-aminopropylfangchinoline (**7**) were prepared. As shown in Table 1, compound **6** gave a satisfactory activity with an inhibition rate of 76.1% while the dephthalimido product **7** gave an obvious drop in activity. It was then assumed that the introduction of some substituent on the amino group of **7** might recover the activity, and consequently a variety of acyl or sulfonyl motives were attached to the amino group and a series of novel fangchinoline derivatives **8a**–**m** were generated. As anticipated, all compounds in this series gave higher activity than **7**, with inhibition rates varying from 35.4% to 87.0%, except for compound **8j**–**m** that failed to give inhibition data due to their high cellular toxicity at the tested concentration of 5 μM. Compound **8a** gave the highest inhibition rate of 87%, however, the introduction of extra electron-donating methyl (**8b**) and methyloxyl groups(**8c** or **8d**) or a electron-withdrawing nitro (**8e**) group on the benzene ring caused significant decreases in activity. Additionally, the replacement of the benzene ring by bioisosteric pyridine (**8g**) or furan (**8h**) or saturated morphine (**8i**) rings also gave reduced activity to varying degrees.

Next, the effects of all the target compounds on IL-1β release were further confirmed at a protein level by western blot assays, as shown in Figure 2A. The addition of LPS and NIG significantly induced IL-1β release, while the addition of **3b**, **3d**, **3e**, **3g**, **5a**, **5b**, **6**, **8a 8f** or **8i** reversed the increase almost completely, which was consistent with the ELISA results. Next, compounds exhibiting an inhibitory rate of over 60% in Table 1 and a significantly reducing effect on IL-1β level in Figure 2A were tested to evaluate their cytotoxic effects at the given concentration of 5 μM in THP-1 cells. MTT assays revealed that compounds **3b**, **3e**, **3g**, **3h**, **5a**, **6**, **8a**, **8f** and **8i** maintained high cell proliferations of over 75%, indicating their low cytotoxicity at the given concentration. Compounds **3d**, **3f**, **4e**, **5b** caused massive cell death and thus were excluded from further study. Compounds **3h**, **5a**, **6**, and **8a** were chose as the representative compounds for the subsequent experiments based on their chemical structure.

We further examined the ability of the representative compounds **3h**, **5a**, **6**, **8a** to attenuate inflammation by testing the IC_50_ inhibition values on IL-1β release. As shown in Figure 3A, these compounds suppressed IL-1β release in a dose-dependent manner, and gave IC_50_ values of 4.53, 6.98, 3.70 and 4.03 μM, respectively. Next their cellular cytotoxicities (CC_50_ values) were further calculated. As shown in Figure 3B, compounds **3h**, **5a**, **6** and **8a** gave CC_50_ values of 19.93, 20.45, 17.62 and 13.66 μM, respectively. Compounds **3h** and **6** gave higher SI (CC_50_/IC_50_) values of 4.4 and 4.8, respectively, and were thus selected to investigate the preliminary mechanism of IL-1β release inhibition.

The effects of **3h** and **6** on the expression of caspase1, IL-1β, NLRP3, pro-IL-1β and pro-caspase1, as key components of NLRP3 inflammasome, were initially evaluated by western blot assay in THP-1 cells. As shown in Figure 4A, LPS/NIG treatment of greatly induced the release of IL-1β and caspase 1, which was suppressed by the addition of **3h** or **6** (5 μM). At the meantime, no significant change in the levels of pro-IL-1β, pro-caspase1 and protein NLRP3 was observed as desired, and the addition of **3h** and **6** (5 μM) did not affect their levels either. Furthermore, the inhibition effects on protein caspase1 of **3h** and **6** (50 or 100 μM) were evaluated by a Caspase1 Activity Assay Kit. As depicted in Figure 4B, neither of them showed any inhibitory effect on caspase1 while the positive control Z-VAD-FMK (25 μM) showed potent inhibitory activity, suggesting neither of them act on caspase1.

The neutrality of **3h** and **6** on the expression of pro-IL-1β and NLRP3 depicted in Figure 4A suggested that these compounds might not impact the NF-κB signaling pathway, since the production of NLRP3 and pro-IL-1β in the cytoplasm is mediated by this pathway [23]. Therefore, the influences of **3h** and **6** on several critical signaling steps involved in NF-κB signaling pathway were evaluated by western blot assay. As shown in Figure 4C, the treatment of tumor necrosis factor-alpha (TNFα), a commonly recognized proinflammatory cytokine induced by LPS [24], successfully activated the NF-κB pathway within 15 minutes. The expressions of TAK1 and P-IKK proteins greatly increased, while the expression of IKBα protein dramatically decreased. However, neither the addition of **3h**-nor **6** could reverse these changes, indicating that these compounds did not act by targeting the NF-κB signaling pathway or the generation of NLRP3 protein.

Then we moved to investigate the MAPK signaling pathway involved in the activation of NLRP3 inflammasome, therein several iconic proteins such as JNK, P38 and ERK, were evaluated by western blot assay. By the stimulation of TNFα, the MAPK pathway was activated, and upregulated phosphorylations of JNK, P38 and ERK were observed, as depicted in Figure 4C. However, both **3h** and **6** hardly exhibited any reversal effects on these expression changes, suggesting that these compounds did not affect the MAPK signaling pathway, and thus might not affect the formation or activation of NLRP3 inflammasome.

LPS/NIG-triggered inflammasome activation features formation of the ASC pyroptosome structure that is believed to mediate caspase-1 activation [25]. Finally, we examined the effects of compound **3h** and **6** on the behavior of NLRP3 and ASC combination using immunofluorescence method. As shown in Figure 5, the LPS/NIG treatment triggered the ASC (labeled as green) pyroptosome structure formation (NLRP3 and ASC combination), while the extra addition of **3h** and **6** greatly abolished the formation of ASC pyroptosome structure. Thus, **3h** and **6** inhibition of LPS/NIG-triggered NLRP3 inflammasome activation likely occurrs via suppression of the NLRP3 and ASC combination, and **6** seemed to display a higher inhibition ability than **3h**.

To further identify the direct target, molecular docking analyses of **6** withNLRP3 and ASC were performed, respectively. The docking models of **6** with NLRP3^pyrin^ (PDB code: 3Q2F), ASC^Pyrin^ (PDB code: 3J63) and ASC^CARD^ (PDB code: 6N1H) were constructed in Discovery Studio 4.5 (BIOVIA, San Diego, CA). As shown in Figure 6A, **6** fitted nicely in the active binding site of NLRP3^pyrin^. Figure 6B depicts the main interactions between **6** and NLRP3. The hydrogen bond with GLN A:45, Van der Waals forces and hydrophobic interactions contributed together to the strong interaction. Libdock program was then applied to quantify these interactions, which gave a libdock score of 107.3, suggesting the strong interaction between **6** and NLRP3. As a comparison, **6** could not fit in ASC^Pyrin^ or ASC^CARD^ (data not shown), indicating its inert interaction with ACS.

## 3. Materials and Methods

### 3.1. Apparatus, Materials, and Analysis Reagents

Unless otherwise noted, all commercial reagents and solvents were obtained from the commercial provider and used without further purification. Melting points (mp) were obtained with a MP90 melting point apparatus and were uncorrected (Mettler-Toledo, Greifensee, Switzerland). ^1^H-NMR and ^13^C-NMR spectra were recorded on a Bruker Avance III 500 (500 MHz for ^1^H-NMR; 126 MHz for ^13^C-NMR) spectrometer (Varian, San Francisco, CA, USA) with Me_4_Si as the internal standard. ESI high-resolution mass spectra (HRMS) were recorded on an Autospec Ultima-TOF spectrometer (Micromass UK Ltd., Manchester, UK). Flash chromatography was performed on a Combiflash Rf 200 system (Teledyne-Isco, Lincoln, NE, USA).

### 3.2. Chemistry

#### 3.2.1. General Procedure for the Preparation of **3d** and **3h**

Compound **1** (608 mg, 1 mmol) was dissolved in DMF (8 mL), to which NaH (48 mg, 2 mmol) was added. The mixture was stirred for 30 min at ambient temperature and then an alkyl or pyridyl halide (1.1 or 2 mmol) was added. The mixture was stirred for 1–10 h at ambient temperature until TLC showed the completion of the reaction. After the solvent was condensed, the residue was suspended in water (40 mL) and extracted with DCM (2 × 40 mL). The combined organic phase was washed with water and brine, dried over anhydrous Na_2_SO_4_ and filtered. After removal of the solvent *in vacuo*, the residue was purified over by flash chromatography over silica gel using a gradient of DCM/MeOH as the eluent and achieved compounds **3d** and **3h**.

*7O-Ethoxyethylfangchinoline* (**3d**) 197 mg, 29% yield; light yellow solid; mp: 116–118 °C. ^1^H-NMR (CDCl_3_) δ 7.36 (dd, *J* = 8.2, 2.1 Hz, 1H), 7.15 (dd, *J* = 8.1, 2.5 Hz, 1H), 6.90 (d, *J* = 8.0 Hz, 1H), 6.86 (d, *J* = 8.1 Hz, 1H), 6.81 (dd, *J* = 8.3, 2.5 Hz, 1H), 6.53 (d, *J* = 1.7 Hz, 1H), 6.50 (s, 1H), 6.32–6.29 (m, 2H), 5.98 (s, 1H), 3.93 (s, 3H), 3.84–3.79 (m, 1H), 3.78–3.73 (m, 4H), 3.70–3.65 (m, 1H), 3.57–3.44 (m, 3H), 3.40–3.34 (m, 5H), 3.33–3.28 (m, 1H), 3.16–3.06 (m, 2H), 2.98–2.77 (m, 5H), 2.77–2.68 (m, 2H), 2.63 (s, 3H), 2.54–2.42 (m, 2H), 2.34 (s, 3H), 1.13 (t, *J* = 7.0 Hz, 3H); ^13^C-NMR (CDCl_3_) δ 153.8, 151.5, 149.5, 148.8, 148.5, 147.2, 143.9, 136.7, 135.1, 134.7, 132.8, 130.3, 128.2, 128.1, 127.4, 122.9, 122.8, 122.1 (2), 120.3, 116.1, 112.7, 111.6, 105.8, 71.5, 69.0, 66.6, 64.4, 61.6, 56.2, 55.9 (2), 45.6, 44.4, 42.7, 42.4, 42.0, 40.3, 24.6, 22.2, 15.3; HRMS: calcd for C_41_H_49_N_2_O_7_ [M + H]^+^: 681.3534, found: 681.3540.

*7O-2″-Methyl-6″-methylpyridinefangchinoline* (**3h**) 278 mg, 39% yield; light yellow solid; mp: 156–158 °C. ^1^H-NMR (CDCl_3_) δ 7.47 (t, *J* = 7.7 Hz, 1H), 7.29 (dd, *J* = 8.2, 2.1 Hz, 1H), 7.13 (dd, *J* = 8.1, 2.5 Hz, 1H), 6.95 (d, *J* = 7.6 Hz, 1H), 6.92 (d, *J* = 8.2 Hz, 1H), 6.87 (d, *J* = 8.2 Hz, 1H), 6.82–6.77 (m, 2H), 6.53 (d, *J* = 1.8 Hz, 1H), 6.52 (s, 1H), 6.34 (s, 1H), 6.26 (dd, *J* = 8.3, 2.1 Hz, 1H), 5.91 (s, 1H), 4.72 (d, *J* = 13.1 Hz, 1H), 4.39 (d, *J* = 13.1 Hz, 1H), 3.93 (s, 3H), 3.82–3.77 (m, 1H), 3.70 (s, 3H), 3.60–3.51 (m, 2H), 3.45–3.39 (m, 4H), 3.28–3.22 (m, 1H), 3.00–2.91 (m, 2H), 2.83–2.66 (m, 5H), 2.57–2.45 (m, 5H), 2.36 (s, 3H), 2.29 (s, 3H); ^13^C-NMR (CDCl_3_) δ 157.4, 156.8, 153.8, 151.4, 149.3, 148.6, 148.5, 147.1, 144.0, 136.7, 136.5, 134.7, 134.6, 132.6, 130.2, 128.3, 127.9, 127.4, 123.0, 122.8, 121.9, 121.9, 121.4, 120.5, 117.6, 116.1, 112.8, 111.5, 105.9, 74.2, 63.8, 61.6, 56.1, 55.9, 55.7, 45.4, 44.3, 42.4, 42.1, 41.9, 39.5, 24.7, 24.3, 22.1; HRMS: calcd for C44H48N3O6 [M + H]^+^: 714.3538, found: 714.3528.

#### 3.2.2. General Procedure for the Preparation of **4c–e**, **5a** and **5b**

To a solution of compound **1** (608 mg, 1 mmol) in DCM (8 mL), TEA (152 mg, 1.5 mmol) was added. The mixture was stirred for 30 min at ambient temperature, then the appropriate acyl or sulfonyl chloride (1.1 mmol) was added. The mixture was stirred for 1–5 h at ambient temperature until TLC indicated the completion of the reaction. The reaction mixture was diluted with water and extracted with DCM (2 × 30 mL). The combined organic phase was washed with water and brine, dried over anhydrous Na_2_SO_4_ and filtered, followed by solvent removal. The residue was purified over by flash chromatography over silica gel using a DCM/MeOH gradient as the eluent, to give compounds **4c**–**4e**, **5a** or **5b**.

*7O-Valerylfangchinoline* (**4c**) 353 mg, 51% yield; light yellow solid; mp: 203–205 °C. ^1^H-NMR (CDCl_3_) δ 7.33 (dd, *J* = 8.2, 2.1 Hz, 1H), 7.13 (dd, *J* = 8.1, 2.5 Hz, 1H), 6.92–6.84 (m, 2H), 6.79 (dd, *J* = 8.3, 2.4 Hz, 1H), 6.52 (s, 1H), 6.48 (s, 1H), 6.34 (s, 1H), 6.26 (d, *J* = 7.0 Hz, 1H), 5.94 (s, 1H), 3.92 (s, 3H), 3.81–3.74 (m, 2H), 3.70 (s, 3H), 3.60–3.52 (m, 1H), 3.51–3.43 (m, 1H), 3.41 (s, 3H), 3.30–3.23 (m, 1H), 2.99–2.70 (m, 7H), 2.60 (s, 3H), 2.56–2.46 (m, 2H), 2.35 (s, 3H), 1.86–1.73 (m, 2H), 1.43–1.35 (m, 2H), 1.29–1.21 (m, 2H), 0.87 (t, *J* = 7.3 Hz, 3H); ^13^C-NMR (CDCl_3_) δ 170.3, 153.3, 149.9, 149.4, 148.7, 147.2, 147.1, 142.8, 134.9, 132.5 (2), 130.7, 130.2, 128.9, 128.2, 127.9, 122.8 (2), 122.0 (2), 120.5, 116.2, 112.6, 111.6, 105.6, 64.2, 61.4, 56.2, 56.0, 55.7, 45.7, 44.1, 42.7, 42.3, 41.6, 39.8, 32.7, 26.6, 25.1, 22.2, 22.1, 13.8; HRMS: calcd for C_42_H_49_N_2_O_7_ [M + H]^+^: 693.3534, found: 693.3528.

*7O-Isobutyrylfangchinoline* (**4d**) 353 mg, 52% yield; light orange solid; mp: 256–258 °C. ^1^H-NMR (CDCl_3_) δ 7.33 (dd, *J* = 8.2, 2.1 Hz, 1H), 7.13 (dd, *J* = 8.1, 2.5 Hz, 1H), 6.91–6.83 (m, 2H), 6.79 (dd, *J* = 8.3, 2.5 Hz, 1H), 6.51 (s, 1H), 6.48 (s, 1H), 6.34 (s, 1H), 6.27 (d, *J* = 7.3 Hz, 1H), 5.95 (s, 1H), 3.92 (s, 3H), 3.80–3.72 (m, 2H), 3.69 (s, 3H), 3.60–3.46 (m, 2H), 3.40 (s, 3H), 3.31–3.25 (m, 1H), 2.99–2.69 (m, 7H), 2.60 (s, 3H), 2.57–2.45 (m, 2H), 2.35 (s, 3H), 2.00–1.92 (m, 1H), 1.01–0.89 (m, 6H); ^13^C-NMR (CDCl_3_) δ 170.3, 153.8, 149.8, 149.3, 148.5, 147.3, 147.0, 143.0, 134.8, 134.6, 132.5 (2), 130.2 (2), 128.8, 128.5, 122.8, 122.0 (4), 116.2, 111.6 (2), 105.80, 64.3, 61.4, 56.1, 56.0, 55.8, 45.9, 44.1, 42.6, 42.4, 41.6, 40.8, 33.2, 24.8, 22.3, 18.8, 18.7; HRMS: calcd for C_41_H_47_N_2_O_7_ [M + H]^+^: 679.3378, found: 679.3390.

*7O-3″,3″-Dimethylbutyrylfangchinoline* (**4e**) 247 mg, 35% yield; light yellow solid; mp: 164–166 °C. ^1^H-NMR (CDCl_3_) δ 7.34 (dd, *J* = 8.2, 1.9 Hz, 1H), 7.13 (dd, *J* = 8.1, 2.5 Hz, 1H), 6.87 (dd, *J* = 18.4, 8.2 Hz, 2H), 6.79 (dd, *J* = 8.2, 2.0 Hz, 1H), 6.51 (s, 1H), 6.49 (s, 1H), 6.34 (s, 1H), 6.26 (d, *J* = 7.5 Hz, 1H), 5.94 (s, 1H), 3.92 (s, 3H), 3.78–3.71 (m, 2H), 3.70 (s, 3H), 3.58–3.50 (m, 2H), 3.40 (s, 3H), 3.31–3.24 (m, 1H), 2.99–2.67 (m, 7H), 2.59 (s, 3H), 2.56–2.41 (m, 2H), 2.35 (s, 3H), 1.74–1.69 (m, 1H), 1.56–1.48 (m, 1H), 0.94 (s, 9H); ^13^C-NMR (CDCl_3_) δ 170.3, 153.7, 149.8, 149.3, 148.6, 147.1, 147.0, 142.8, 134.9, 134.7, 132.5 (2), 130.7, 130.3, 128.9 (2), 127.5, 122.8 (2), 122.0 (2), 116.1, 112.4, 111.4, 105.5, 64.3, 61.2, 56.1(2), 55.7, 46.8, 45.7, 44.0, 42.7, 42.3, 41.5, 30.5, 29.3 (4), 24.2, 22.1; HRMS: calcd for C_43_H_51_N_2_O_7_ [M + H]^+^: 707.3691, found: 679.3691.

*7O-1″-Propanesulfonylfangchinoline* (**5a**) 243 mg, 34% yield; brown solid; mp: 228–230 °C. ^1^H-NMR (CDCl_3_) δ 7.37 (dd, *J* = 8.2, 2.1 Hz, 1H), 7.15 (dd, *J* = 8.2, 2.5 Hz, 1H), 6.87–6.79 (m, 3H), 6.54 (s, 1H), 6.49 (d, *J* = 1.2 Hz, 1H), 6.38 (s, 1H), 6.34 (dd, *J* = 8.3, 2.1 Hz, 1H), 6.04 (s, 1H), 3.93 (s, 3H), 3.90–3.84 (m, 1H), 3.78 (s, 3H), 3.73–3.68 (m, 1H), 3.55–3.47 (m, 2H), 3.37 (s, 3H), 3.35–3.30 (m, 1H), 3.01–2.85 (m, 6H), 2.84–2.66 (m, 3H), 2.64–2.57 (m, 4H), 2.50–2.42 (m, 1H), 2.29 (s, 3H), 1.85–1.69 (m, 2H), 0.98 (t, *J* = 7.5 Hz, 3H); ^13^C-NMR (CDCl_3_) δ 154.2, 150.8, 150.0, 149.4, 148.7, 147.4, 142.8, 135.4, 135.0, 133.0, 132.9, 130.9, 128.7, 128.3, 128.1, 124.1, 123.3, 122.7, 122.5, 122.1, 116.3, 113.0, 111.9, 106.6, 64.6, 61.9, 56.6 (2), 56.4, 54.0, 45.7, 44.2, 42.9, 42.8, 42.0, 39.7, 25.0, 22.6, 17.7, 13.4; HRMS: calcd for C_40_H_47_N_2_O_8_S [M + H]^+^: 715.3048, found: 715.3048.

*7O-m-Chlorobenzenesulfonylfangchinoline* (**5b**) 392 mg, 50% yield; light yellow solid; mp: 166–168 °C. ^1^H-NMR (CDCl_3_) δ 7.72 (t, *J* = 1.8 Hz, 1H), 7.64–7.61 (m, 1H), 7.56–7.53 (m, 1H), 7.40 (t, *J* = 8.0 Hz, 1H), 7.36 (dd, *J* = 8.2, 2.1 Hz, 1H), 7.18 (dd, *J* = 8.1, 2.5 Hz, 1H), 6.89–6.85 (m, 2H), 6.75 (dd, *J* = 8.3, 2.5 Hz, 1H), 6.47 (d, *J* = 8.2 Hz, 2H), 6.31 (s, 1H), 6.26 (dd, *J* = 8.3, 2.1 Hz, 1H), 5.19 (s, 1H), 3.94 (s, 3H), 3.80–3.75 (m, 1H), 3.68–3.65 (m, 1H), 3.56–3.47 (m, 5H), 3.33–3.28 (m, 4H), 2.99–2.89 (m, 4H), 2.80–2.72 (m, 2H), 2.71–2.65 (m, 4H), 2.57–2.51 (m, 1H), 2.49–2.42 (m, 1H), 2.29 (s, 3H); ^13^C-NMR (CDCl_3_) δ 154.0, 151.0, 149.4, 148.8, 147.9, 147.3, 142.5, 139.2, 134.9, 134.6, 133.8, 132.9, 132.7, 130.4, 129.9, 128.7 (2), 128.3 (2), 126.9, 124.0, 122.9, 122.0, 121.8 (2), 121.0, 116.3, 112.8, 111.6, 106.2, 64.0, 61.5, 56.2, 55.9, 55.8, 45.1, 43.9, 42.4 (2), 42.1, 39.7, 24.3, 22.2; HRMS: calcd for C_43_H_44_ClN_2_O_8_S [M + H]^+^: 783.2501, found: 783.2499.

#### 3.2.3. Procedure for the Preparation of 7O-Phthalimidepropylfangchinoline (**6**)

To a solution of compound **1** (6.0 g, 9.8 mmol) in DMF (78 mL), NaH (0.47 g, 19.6 mmol) was added. The mixture was stirred for 30 min at ambient temperature, then *N*-(3-bromopropyl)-phthalimide (3.2 g, 11.8 mmol) was added in an ice-water bath. The mixture was stirred for 30 min in the ice-water bath under N_2_. The temperature gradually raised to room temperature under N_2_ and the mixture was stirred 6 h until the reaction completed. The reaction mixture was diluted with water and extracted with DCM (2 × 80 mL). The combined organic phase was washed with water and brine, dried over anhydrous Na_2_SO_4_ and filtered, followed by solvent removal. The residue was purified over by flash chromatography over silica gel using a gradient of DCM/MeOH as the eluent to givecompound **6** (3.90 g). 50% yield; light yellow solid; mp: 190–192 °C. ^1^H-NMR (CDCl_3_) δ 7.85–7.80 (m, 2H), 7.73–7.68 (m, 2H), 7.38 (dd, *J* = 8.2, 2.1 Hz, 1H), 7.14 (dd, *J* = 8.2, 2.5 Hz, 1H), 6.90–6.84 (m, 2H), 6.82 (dd, *J* = 8.3, 2.5 Hz, 1H), 6.54 (s, 1H), 6.50 (d, *J* = 1.2 Hz, 1H), 6.33 (dd, *J* = 8.3, 2.1 Hz, 1H), 6.29 (s, 1H), 5.98 (s, 1H), 3.93 (s, 3H), 3.88–3.83 (m, 1H), 3.75–3.70 (m, 4H), 3.61–3.41 (m, 6H), 3.38 (s, 3H), 3.35–3.28 (m, 1H), 2.96–2.87 (m, 4H), 2.81–2.67 (m, 4H), 2.54–2.47 (m, 1H), 2.45 (s, 3H), 2.32 (s, 3H), 1.66–1.57 (m, 1H), 1.50–1.41 (m, 1H); ^13^C-NMR (CDCl_3_) δ 168.2 (2), 153.7, 151.4, 149.4, 148.7, 148.2, 147.0, 143.7, 136.5, 134.9, 134.5, 133.9 (2), 132.6, 132.1 (2), 130.3, 128.2, 127.8, 123.1 (2), 122.8, 122.5, 122.0 (2), 120.0 (2), 115.9, 112.8, 111.4, 105.8, 70.2, 64.0, 61.4, 56.1, 55.8, 55.7, 45.2, 44.2, 42.3, 42.1, 41.7, 39.7, 35.3, 28.4, 24.3, 22.1; HRMS: calcd for C_48_H_50_N_3_O_8_ [M + H]^+^: 796.3592, found: 796.3604.

#### 3.2.4. Procedure for the Preparation of 7*O*-Aminopropylfangchinoline (**7**)

Compound **6** (0.6 g, 0.7 mmol) was dissolved in 15 mL of EtOH, to which 80% hydrazine hydrate (0.88 g, 14 mmol) was added. The mixture was stirred overnight at ambient temperature. After filtration, the filtrate was evaporated, suspended with water (20 mL) and extracted with DCM (2 × 20 mL). The combined organic phase was washed with water and brine, dried over anhydrous MgSO_4_ and filtered. The filtrate was concentrated to give product **7** (0.45 g). Yield: 90%; light yellow solid; mp: 126–128 °C. ^1^H-NMR (CDCl_3_) δ 7.35 (dd, *J* = 8.2, 2.1 Hz, 1H), 7.14 (dd, *J* = 8.1, 2.5 Hz, 1H), 6.89 – 6.84 (m, 2H), 6.81 (dd, *J* = 8.3, 2.5 Hz, 1H), 6.54–6.51 (m, 2H), 6.32 (dd, *J* = 8.3, 2.1 Hz, 1H), 6.30 (s, 1H), 5.98 (s, 1H), 3.93 (s, 3H), 3.86–3.79 (m, 1H), 3.76–3.69 (m, 4H), 3.63–3.57 (m, 1H), 3.56–3.48 (m, 1H), 3.46–3.40 (m, 1H), 3.40–3.33 (m, 4H), 3.28–3.21 (m, 1H), 2.97–2.66 (m, 7H), 2.61 (s, 3H), 2.55–2.37 (m, 4H), 2.32 (s, 3H), 1.78 (br s, 2H), 1.33–1.20 (m, 2H); ^13^C-NMR (DMSO-*d_6_*) δ 152.8, 150.9, 148.8, 147.8, 147.7, 146.5, 143.0, 136.3, 135.6, 134.1, 132.5, 130.5, 128.3, 128.2, 127.5, 122.8, 122.1, 121.5, 121.2, 119.6, 115.1, 112.6, 111.8, 106.2, 70.0, 62.9, 61.0, 55.6, 55.5 (2), 44.9, 43.1, 42.2, 41.9, 41.3, 38.3, 37.6, 32.8, 24.8, 21.0. HRMS: calcd for C_40_H_48_N_3_O_6_ [M + H]^+^: 666.3538, found: 666.3534.

#### 3.2.5. General Procedure for the Preparation of **8a**–**m**

To a solution of compound **7** (400 mg, 0.6 mmol) in DCM (10 mL), TEA (121 mg, 1.2 mmol) and the appropriate acyl or sulfonyl halide (0.72 mmol) was added in an ice-water bath. The mixture was stirred for 30 min in the ice-water bath under N_2_. The reaction temperature was then allowed to gradually rise to room temperature and the mixture was stirred 0.5–6 h until the reaction was complete. The mixture was diluted with water and extracted with DCM (2 × 30 mL). The combined organic phase was washed with water and brine, dried over anhydrous Na_2_SO_4_ and filtered, followed by solvent removal. The residue was purified over by flash chromatography over silica gel using a DCM/MeOH gradient as eluent, giving target compounds **8a**–**m**.

*7O-Benzamidepropylfangchinoline* (**8a**) 328 mg, 71% yield; light yellow solid; mp: 148–150 °C. ^1^H-NMR (DMSO-*d*_6_) δ 8.26 (t, *J* = 5.6 Hz, 1H), 7.81–7.78 (m, 2H), 7.54–7.50 (m, 1H), 7.48–7.44 (m, 2H), 7.36 (dd, *J* = 8.2, 2.1 Hz, 1H), 7.04 (dd, *J* = 8.2, 2.6 Hz, 1H), 6.92 (d, *J* = 8.3 Hz, 1H), 6.77 (dd, *J* = 8.2, 1.8 Hz, 1H), 6.68 (dd, *J* = 8.2, 2.6 Hz, 1H), 6.60 (s, 1H), 6.40 (s, 1H), 6.35–6.31 (m, 2H), 5.90 (s, 1H), 3.83–3.78 (m, 4H), 3.65 (s, 3H), 3.62–3.57 (m, 1H), 3.51–3.47 (m, 1H), 3.47–3.39 (m, 1H), 3.34–3.30 (m, 1H), 3.29 (s, 3H), 3.23–3.18 (m, 1H), 3.16–3.08 (m, 2H), 3.04–2.97 (m, 1H), 2.87–2.60 (m, 7H), 2.38 (s, 3H), 2.37–2.28 (m, 2H), 2.17 (s, 3H), 1.33–1.23 (m, 2H); ^13^C-NMR (DMSO-*d*_6_) δ 166.7, 153.5, 151.6, 149.6, 148.5, 148.5, 147.3, 143.8, 137.0, 136.3, 135.4, 134.8, 133.3, 131.7, 131.2, 129.1, 129.0 (2), 128.5, 127.7 (2), 123.6, 123.0, 122.2, 122.0, 120.4, 115.9, 113.4, 112.5, 107.0, 70.9, 63.5, 61.8, 56.3, 56.3, 55.6, 45.5, 43.8, 42.7, 42.6, 42.1, 39.0, 37.0, 29.9, 25.1, 21.7; HRMS: calcd for C_47_H_52_N_3_O_7_ [M + H]^+^: 770.3800, found: 770.3799.

*7O-p-Methylbenzamidepropylfangchinoline* (**8b**) 165 mg, 35% yield; light yellow solid; mp: 130–132 °C. ^1^H-NMR (DMSO-*d*_6_) δ 8.17 (t, *J* = 5.6 Hz, 1H), 7.71–7.68 (m, 2H), 7.36 (dd, *J* = 8.2, 2.1 Hz, 1H), 7.27 (s, 1H), 7.25 (s, 1H), 7.03 (dd, *J* = 8.2, 2.6 Hz, 1H), 6.92 (d, *J* = 8.3 Hz, 1H), 6.77 (dd, *J* = 8.2, 1.8 Hz, 1H), 6.68 (dd, *J* = 8.2, 2.6 Hz, 1H), 6.59 (s, 1H), 6.40 (s, 1H), 6.34–6.31 (m, 2H), 5.89 (s, 1H), 3.83–3.78 (m, 4H), 3.65 (s, 3H), 3.61–3.55 (m, 1H), 3.51–3.47 (m, 1H), 3.46–3.38 (m, 1H), 3.32–3.28 (m, 4H), 3.22–3.17 (m, 1H), 3.16–3.06 (m, 2H), 3.03–2.94 (m, 1H), 2.87–2.77 (m, 3H), 2.75–2.60 (m, 4H), 2.41–2.36 (m, 4H), 2.34 (s, 3H), 2.32–2.28 (m, 1H), 2.17 (s, 3H), 1.31–1.22 (m, 2H). ^13^C-NMR (DMSO-*d*_6_) δ 165.9, 152.9, 151.0, 149.0, 148.0, 147.9, 146.7, 143.2, 140.9, 136.4, 136.0, 134.2, 132.6, 132.0, 130.5, 128.8 (2), 128.4, 128.3, 127.8, 127.1 (2), 123.0, 122.3, 122.0, 121.4, 119.7, 115.3, 112.8, 111.9, 106.3, 70.4, 62.9, 61.2, 59.8, 55.7 (2), 44.9, 43.3, 42.1, 42.0, 41.5, 38.4, 36.4, 29.3, 25.2, 24.5, 21.0. HRMS: calcd for C_48_H_54_N_3_O_7_ [M + H]^+^: 784.3956, found: 784.3955.

*7O-p-Methoxybenzamidepropylfangchinoline* (**8c**) 211 mg, 44% yield; light yellow solid; mp: 143–145 °C. ^1^H-NMR (DMSO-*d*_6_) δ 8.11 (t, *J* = 5.6 Hz, 1H), 7.79–7.75 (m, 2H), 7.36 (dd, *J* = 8.2, 2.1 Hz, 1H), 7.03 (dd, *J* = 8.2, 2.6 Hz, 1H), 7.01–6.97 (m, 2H), 6.92 (d, *J* = 8.2 Hz, 1H), 6.77 (dd, *J* = 8.2, 1.8 Hz, 1H), 6.68 (dd, *J* = 8.2, 2.6 Hz, 1H), 6.59 (s, 1H), 6.39 (s, 1H), 6.34–6.31 (m, 2H), 5.89 (s, 1H), 3.83–3.77 (m, 7H), 3.65 (s, 3H), 3.61–3.56 (m, 1H), 3.50–3.47 (m, 1H), 3.46–3.38 (m, 1H), 3.33–3.30 (m, 1H), 3.29 (s, 3H), 3.22–3.16 (m, 1H), 3.16–3.06 (m, 2H), 3.02–2.95 (m, 1H), 2.87–2.60 (m, 7H), 2.39 (s, 3H), 2.37–2.28 (m, 2H), 2.17 (s, 3H), 1.30–1.22 (m, 2H); ^13^C-NMR (DMSO-*d*_6_) δ 165.4, 161.3, 152.7, 150.9, 148.8, 147.8, 147.7, 146.5, 143.0, 136.2, 135.5, 134.1, 132.5, 130.4, 128.7 (2), 128.3, 128.2, 127.7, 126.8, 122.8, 122.2, 121.4, 121.2, 119.6, 115.1, 113.3 (2), 112.7, 111.8, 106.2, 70.2, 62.7, 61.0, 55.6, 55.5 (2), 55.2, 44.8, 43.1, 42.0, 41.8, 41.3, 38.4, 36.2, 29.2, 24.3, 21.0; HRMS: calcd for C_48_H_54_N_3_O_8_ [M + H]^+^: 800.39054, found:800.3908.

*7O-3″,4″,5″-Trimethoxybenzamidepropylfangchinoline* (**8d**) 222 mg, 43% yield; white solid; mp: 144–146 °C. ^1^H-NMR (DMSO-*d*_6_) δ 8.24 (t, *J* = 5.6 Hz, 1H), 7.30 (dd, *J* = 8.2, 2.1 Hz, 1H), 7.14 (s, 2H), 7.01 (dd, *J* = 8.2, 2.6 Hz, 1H), 6.91 (d, *J* = 8.3 Hz, 1H), 6.76 (dd, *J* = 8.2, 1.8 Hz, 1H), 6.68 (dd, *J* = 8.2, 2.6 Hz, 1H), 6.58 (s, 1H), 6.40 (s, 1H), 6.34–6.30 (m, 2H), 5.89 (s, 1H), 3.82–3.75 (m, 10H), 3.69 (s, 3H), 3.65 (s, 3H), 3.62–3.57 (m, 1H), 3.51–3.47 (m, 1H), 3.45–3.38 (m, 1H), 3.32–3.29 (m, 1H), 3.28 (s, 3H), 3.23–3.08 (m, 3H), 3.04–2.96 (m, 1H), 2.86–2.60 (m, 7H), 2.41 (s, 3H), 2.38–2.26 (m, 2H), 2.16 (s, 3H), 1.33–1.25 (m, 2H); ^13^C-NMR (DMSO-*d*_6_) δ 165.2, 152.7, 152.4 (2), 150.9, 148.8, 147.8, 147.6, 146.5, 143.0, 139.6, 136.3, 135.5, 134.0, 132.9, 130.4, 129.8, 128.3, 128.2, 127.7, 122.8, 122.2, 121.4, 121.2, 119.5, 115.1, 112.7, 111.8, 106.2, 104.4 (2), 70.3, 62.7, 61.0, 60.0, 55.8 (2), 55.5 (3), 44.8, 43.1, 42.0, 41.9, 41.3, 38.5, 36.3, 29.2, 24.3, 21.0; HRMS: calcd for C_50_H_58_N_3_O_10_ [M + H]^+^: 860.4117, found: 860.4113.

*7O-p-Nitrobenzamidepropylfangchinoline* (**8e**) 220 mg, 45% yield; orange solid; mp: 148–150 °C. ^1^H-NMR (DMSO-*d*_6_) δ 8.63 (t, *J* = 5.6 Hz, 1H), 8.35–8.31 (m, 2H), 8.04–8.01 (m, 2H), 7.40 (dd, *J* = 8.2, 2.1 Hz, 1H), 7.05 (dd, *J* = 8.2, 2.6 Hz, 1H), 6.92 (d, *J* = 8.3 Hz, 1H), 6.76 (dd, *J* = 8.2, 1.8 Hz, 1H), 6.69 (dd, *J* = 8.2, 2.6 Hz, 1H), 6.61 (s, 1H), 6.40 (s, 1H), 6.33 (dd, *J* = 8.6, 2.0 Hz, 2H), 5.91 (s, 1H), 3.84–3.79 (m, 4H), 3.65 (s, 3H), 3.62–3.56 (m, 1H), 3.51–3.47 (m, 1H), 3.47–3.39 (m, 1H), 3.33–3.27 (m, 4H), 3.24–3.18 (m, 1H), 3.17–3.09 (m, 2H), 3.07–2.99 (m, 1H), 2.86–2.61 (m, 7H), 2.38 (s, 3H), 2.37–2.27 (m, 2H), 2.17 (s, 3H), 1.35–1.24 (m, 2H); ^13^C-NMR (DMSO-*d*_6_) δ 164.3, 152.8, 150.8, 148.8 (2), 147.7, 146.5, 143.0, 140.2, 136.2, 135.5, 134.0, 132.5, 130.4, 128.5 (2), 128.3, 128.2, 127.7, 123.5 (2), 122.8, 122.2, 121.5, 121.2, 119.6, 115.1, 112.7, 111.8, 106.2, 69.9, 62.8, 61.0, 55.6, 55.5 (2), 44.8, 43.1, 42.0, 41.9, 41.3, 38.3, 36.5, 28.9, 24.4, 21.0; HRMS: calcd for C_47_H_51_N_4_O_9_ [M + H]^+^: 815.3659, found: 760.3659.

*7O-p-Chlorobenzamidepropylfangchinoline* (**8f**) 198 mg, 41% yield; light yellow solid; mp: 145–147 °C. ^1^H-NMR (DMSO-*d*_6_) δ 8.36 (t, *J* = 5.6 Hz, 1H), 7.84–7.79 (m, 2H), 7.58–7.52 (m, 2H), 7.37 (dd, *J* = 8.2, 2.1 Hz, 1H), 7.05 (dd, *J* = 8.2, 2.6 Hz, 1H), 6.94–6.90 (m, 1H), 6.77 (dd, *J* = 8.2, 1.8 Hz, 1H), 6.68 (dd, *J* = 8.2, 2.6 Hz, 1H), 6.60 (s, 1H), 6.40 (s, 1H), 6.35–6.31 (m, 2H), 5.90 (s, 1H), 3.83–3.78 (m, 4H), 3.65 (s, 3H), 3.61–3.56 (m, 1H), 3.51–3.47 (m, 1H), 3.47–3.39 (m, 1H), 3.34–3.30 (m, 1H), 3.29 (s, 3H), 3.22–3.17 (m, 1H), 3.16–3.07 (m, 2H), 3.03–2.96 (m, 1H), 2.87–2.60 (m, 7H), 2.38 (s, 3H), 2.36–2.28 (m, 2H), 2.17 (s, 3H), 1.32–1.22 (m, 2H); ^13^C-NMR (DMSO-*d*_6_) δ 165.0, 152.9, 151.0, 149.0, 147.9, 147.8, 146.7, 143.2, 136.4, 135.9, 135.6, 134.2, 133.4, 132.6, 130.5, 129.0 (2), 128.4, 128.4 (2), 128.3, 127.8, 123.0, 122.3, 121.6, 121.4, 119.7, 115.3, 112.8, 111.9, 106.4, 70.2, 62.9, 61.1, 55.7, 55.7 (2), 44.9, 43.2, 42.1, 42.0, 41.5, 38.5, 36.5, 29.1, 24.5, 21.1; HRMS: calcd for C_47_H_51_ClN_3_O_7_ [M + H]^+^: 804.3410, found: 803.3411.

*7O-o-Picolinamidepropylfangchinoline* (**8g**) 102 mg, 22% yield; light yellow solid; mp: 146–148 °C. ^1^H-NMR (DMSO-*d*_6_) δ 8.64 (d, *J* = 4.6 Hz, 1H), 8.58 (t, *J* = 6.0 Hz, 1H), 8.08–7.99 (m, 2H), 7.65–7.59 (m, 1H), 7.32 (dd, *J* = 8.2, 1.7 Hz, 1H), 7.06 (dd, *J* = 8.1, 2.4 Hz, 1H), 6.92 (d, *J* = 8.2 Hz, 1H), 6.77 (dd, *J* = 8.2, 1.2 Hz, 1H), 6.69 (dd, *J* = 8.2, 2.4 Hz, 1H), 6.60 (s, 1H), 6.40 (s, 1H), 6.35–6.30 (m, 2H), 5.90 (s, 1H), 3.81 (s, 3H), 3.79–3.73 (m, 1H), 3.67 (s, 3H), 3.63–3.56 (m, 1H), 3.53–3.39 (m, 2H), 3.33–3.25 (m, 4H), 3.19–3.01 (m, 4H), 2.88–2.60 (m, 7H), 2.42–2.27 (m, 5H), 2.16 (s, 3H), 1.27–1.20 (m, 2H); ^13^C-NMR (DMSO-*d*_6_) δ 163.7, 153.1, 151.2, 150.2, 149.1, 148.6, 148.1, 148.0, 146.8, 143.3, 138.0, 136.4, 135.8, 134.4, 132.8, 130.7, 128.7, 128.5, 128.0, 126.7, 123.1, 122.4, 122.0, 121.8, 121.6, 119.9, 115.5, 112.9, 112.1, 106.4, 70.3, 63.1, 61.3, 55.8 (2), 55.2, 45.1, 43.4, 42.3, 42.2, 41.6, 38.5, 36.2, 29.3, 24.7, 21.3; HRMS: calcd for C_46_H_51_N_4_O_7_ [M + H]^+^: 771.3752, found: 771.3751.

*7O-o-Furamidepropylfangchinoline* (**8h**) 100 mg, 22% yield; light yellow solid; mp: 142–144 °C. ^1^H-NMR (DMSO-*d*_6_) δ 8.11 (t, *J* = 5.8 Hz, 1H), 7.84 (dd, *J* = 1.7, 0.7 Hz, 1H), 7.40 (dd, *J* = 8.2, 2.1 Hz, 1H), 7.08–7.04 (m, 2H), 6.92 (d, *J* = 8.3 Hz, 1H), 6.77 (dd, *J* = 8.2, 1.8 Hz, 1H), 6.69 (dd, *J* = 8.2, 2.6 Hz, 1H), 6.63 (dd, *J* = 3.4, 1.8 Hz, 1H), 6.59 (s, 1H), 6.39 (s, 1H), 6.36–6.31 (m, 2H), 5.90 (s, 1H), 3.83–3.78 (m, 4H), 3.65 (s, 3H), 3.58–3.52 (m, 1H), 3.51–3.46 (m, 1H), 3.46–3.38 (m, 1H), 3.32–3.27 (m, 4H), 3.19–3.11 (m, 2H), 3.09–3.01 (m, 1H), 3.00–2.91 (m, 1H), 2.87–2.60 (m, 7H), 2.41 (s, 3H), 2.38–2.26 (m, 2H), 2.17 (s, 3H), 1.24–1.19 (m, 2H); ^13^C-NMR (DMSO-*d*_6_) δ 157.4, 152.8, 150.8, 148.8, 148.0, 147.7 (2), 146.5, 144.7, 143.0, 136.1, 135.5, 134.1, 132.5, 130.4, 128.3, 128.2, 127.7, 122.8, 122.1, 121.5, 121.2, 119.6, 115.1, 113.0, 112.6, 111.8, 111.7, 106.2, 70.1, 62.8, 61.0, 55.5 (3), 44.8, 43.1, 42.0, 41.8, 41.3, 38.1, 35.6, 29.1, 24.4, 21.0; HRMS: calcd for C_45_H_50_N_3_O_8_ [M + H]^+^: 760.3592, found: 760.3591.

*7O-Morpholine-4″-carboxamidepropylfangchinoline* (**8i**) 211 mg, 45% yield; light yellow solid; mp: 152–154 °C. ^1^H-NMR (DMSO-*d*_6_) δ 7.46 (dd, *J* = 8.2, 2.1 Hz, 1H), 7.09 (dd, *J* = 8.2, 2.6 Hz, 1H), 6.91 (d, *J* = 8.3 Hz, 1H), 6.76 (dd, *J* = 8.2, 1.8 Hz, 1H), 6.69 (dd, *J* = 8.2, 2.6 Hz, 1H), 6.60 (s, 1H), 6.39 (s, 1H), 6.36–6.29 (m, 3H), 5.90 (s, 1H), 3.86–3.80 (m, 4H), 3.67 (s, 3H), 3.54–3.41 (m, 7H), 3.29 (s, 3H), 3.22–3.10 (m, 6H), 2.89–2.59 (m, 10H), 2.47 (s, 3H), 2.39–2.25 (m, 2H), 2.16 (s, 3H), 1.24–1.09 (m, 2H); ^13^C-NMR (DMSO-*d*_6_) δ 157.4, 152.8, 150.9, 148.8, 147.7 (2), 146.5, 143.0, 136.2, 135.5, 134.0, 132.5, 130.4, 128.3, 128.1, 127.6, 122.8, 122.1, 121.5, 121.2, 119.6, 115.1, 112.7, 111.8, 106.2, 70.2, 65.8 (2), 62.9, 61.0, 55.6, 55.5 (2), 44.8, 43.7 (2), 43.1, 42.1, 41.8, 41.3, 38.6, 37.1, 29.6, 24.2, 21.0. HRMS: calcd for C_45_H_55_N_4_O_8_ [M + H]^+^: 779.4014, found: 771.4016.

*7O-3″,4″-Dichlorobenzamidepropylfangchinoline* (**8j**) 318 mg, 63% yield; light yellow solid; mp: 145–147 °C. ^1^H-NMR (DMSO-*d*_6_) δ 8.48 (t, *J* = 5.6 Hz, 1H), 8.03 (d, *J* = 1.5 Hz, 1H), 7.8 –7.75 (m, 2H), 7.36 (dd, *J* = 8.2, 2.1 Hz, 1H), 7.05 (dd, *J* = 8.2, 2.6 Hz, 1H), 6.92 (d, *J* = 8.3 Hz, 1H), 6.77 (dd, *J* = 8.2, 1.8 Hz, 1H), 6.68 (dd, *J* = 8.2, 2.6 Hz, 1H), 6.60 (s, 1H), 6.40 (s, 1H), 6.34–6.30 (m, 2H), 5.89 (s, 1H), 3.82–3.76 (m, 4H), 3.65 (s, 3H), 3.61–3.56 (m, 1H), 3.51–3.47 (m, 1H), 3.47–3.39 (m, 1H), 3.34–3.30 (m, 1H), 3.29 (s, 3H), 3.23–3.17 (m, 1H), 3.16–3.07 (m, 2H), 3.05–2.96 (m, 1H), 2.88–2.60 (m, 7H), 2.38 (s, 3H), 2.36–2.27 (m, 2H), 2.17 (s, 3H), 1.33–1.22 (m, 2H); ^13^C-NMR (DMSO-*d*_6_) δ 163.7, 153.0, 151.0, 149.0, 147.9, 147.8, 146.7, 143.2, 136.4, 135.6, 135.0, 134.2, 133.9, 132.7, 131.3, 130.8, 130.5, 129.1, 128.5, 128.3, 127.9, 127.5, 123.0, 122.3, 121.6, 121.4, 119.7, 115.3, 112.8, 111.9, 106.4, 70.1, 62.9, 61.2, 55.7, 55.7 (2), 45.0, 43.2, 42.2, 42.0, 41.5, 38.4, 36.6, 29.0, 24.5, 21.1; HRMS: calcd for C_47_H_51_Cl_2_N_3_O_7_ [M + H]^+^: 839.3099, found: 839.3104.

*7O-Benzenesulfonamidepropylfangchinoline* (**8k**) 102 mg, 21% yield; light yellow solid; mp: 146–148 °C. ^1^H-NMR (DMSO-*d*_6_) δ 7.79–7.73 (m, 2H), 7.67–7.57 (m, 4H), 7.46 (dd, *J* = 8.3, 2.1 Hz, 1H), 7.09 (dd, *J* = 8.2, 2.6 Hz, 1H), 6.91 (d, *J* = 8.3 Hz, 1H), 6.77 (dd, *J* = 8.2, 1.8 Hz, 1H), 6.68 (dd, *J* = 8.2, 2.6 Hz, 1H), 6.57 (s, 1H), 6.37 (s, 1H), 6.35–6.31 (m, 2H), 5.88 (s, 1H), 3.88–3.82 (m, 1H), 3.80 (s, 3H), 3.63 (s, 3H), 3.52–3.40 (m, 3H), 3.31–3.24 (m, 4H), 3.21–3.14 (m, 1H), 3.13–3.05 (m, 1H), 2.86–2.57 (m, 7H), 2.57–2.51 (m, 1H), 2.49–2.42 (m, 4H), 2.37–2.25 (m, 2H), 2.16 (s, 3H), 1.18–1.06 (m, 2H). ^13^C-NMR (DMSO-*d*_6_) δ 152.8, 150.8, 148.8, 147.9, 147.5, 146.5, 143.0, 140.5, 136.1, 135.5, 134.1, 132.5, 132.2, 130.5, 129.1 (2), 128.2 (2), 127.6, 126.3 (2), 122.8, 122.1, 121.5, 121.2, 119.7, 115.1, 112.6, 111.8, 106.1, 69.6, 62.5, 61.0, 55.6, 55.5, 55.5, 44.6, 43.0, 42.0, 41.8 (2), 41.2, 36.6, 29.5, 24.7, 20.9. HRMS: calcd for C_46_H_52_N_3_O_8_S [M + H]^+^: 806.3470, found: 806.3474.

*7O-p-Methoxybenzenesulfonamidepropylfangchinoline* (**8l**) 261 mg, 52% yield; light yellow solid; mp: 146–148 °C. ^1^H-NMR (DMSO-*d*_6_) δ 7.71–7.67 (m, 2H), 7.45 (dd, *J* = 10.2, 4.0 Hz, 2H), 7.14–7.06 (m, 3H), 6.91 (d, *J* = 8.3 Hz, 1H), 6.77 (dd, *J* = 8.2, 1.8 Hz, 1H), 6.68 (dd, *J* = 8.2, 2.6 Hz, 1H), 6.58 (s, 1H), 6.37 (s, 1H), 6.35–6.31 (m, 2H), 5.88 (s, 1H), 3.88–3.82 (m, 4H), 3.80 (s, 3H), 3.63 (s, 3H), 3.50–3.37 (m, 3H), 3.32–3.25 (m, 4H), 3.20–3.14 (m, 1H), 3.11–3.05 (m, 1H), 2.86–2.59 (m, 7H), 2.49–2.39 (m, 5H), 2.37–2.25 (m, 2H), 2.16 (s, 3H), 1.16–1.06 (m, 2H). ^13^C-NMR (DMSO-*d*_6_) δ 161.9, 152.8, 150.8, 148.8, 147.9, 147.5, 146.5, 143.0, 136.1, 135.5, 134.1, 132.5, 132.1, 130.5, 128.4 (2), 128.2, 128.1, 127.6, 122.8, 122.1, 121.5, 121.2, 119.72, 115.2, 114.2 (2), 112.6, 111.8, 106.1, 69.6, 62.5, 61.0, 55.6, 55.5 (3), 44.6, 43.0, 42.1, 41.8 (2), 41.3, 36.7, 29.4, 24.7, 20.9. HRMS: calcd for C_47_H_54_N_3_O_9_S [M + H]^+^: 836.3575, found: 836.3582.

*7O-p-Methylbenzenesulfonamidepropylfangchinoline* (**8m**) 256 mg, 52% yield; light yellow solid; mp: 146–148 °C. ^1^H-NMR (DMSO-*d*_6_) δ 7.64 (d, *J* = 8.2 Hz, 2H), 7.53 (t, *J* = 6.0 Hz, 1H), 7.46 (dd, *J* = 8.2, 2.1 Hz, 1H), 7.40 (d, *J* = 8.0 Hz, 2H), 7.08 (dd, *J* = 8.2, 2.6 Hz, 1H), 6.91 (d, *J* = 8.3 Hz, 1H), 6.77 (dd, *J* = 8.2, 1.8 Hz, 1H), 6.68 (dd, *J* = 8.2, 2.6 Hz, 1H), 6.57 (s, 1H), 6.37 (s, 1H), 6.35–6.31 (m, 2H), 5.88 (s, 1H), 3.89–3.82 (m, 1H), 3.80 (s, 3H), 3.63 (s, 3H), 3.50–3.36 (m, 3H), 3.31–3.24 (m, 4H), 3.21–3.14 (m, 1H), 3.13–3.04 (m, 1H), 2.85–2.58 (m, 7H), 2.48–2.41 (m, 4H), 2.40–2.24 (m, 6H), 2.16 (s, 3H), 1.16–1.05 (m, 2H). ^13^C-NMR (DMSO-*d*_6_) δ 152.8, 150.8, 148.8, 147.9, 147.5, 146.5, 143.0, 142.4, 137.6, 136.1, 135.5, 134.1, 132.5, 130.5, 129.5 (2), 128.23, 128.1, 127.6, 126.3 (2), 122.8, 122.1, 121.5, 121.2, 119.7, 115.1, 112.6, 111.8, 106.1, 69.6, 62.5, 61.0, 55.6, 55.5 (2), 44.6, 43.0, 42.1, 41.8 (2), 41.2, 36.7, 29.4, 24.7, 20.9, 20.9. HRMS: calcd for C_47_H_54_N_3_O_8_S [M + H]^+^: 820.3626, found: 820.3630.

### 3.3. Biology Assay

#### 3.3.1. Reagents

Anti-IL-1β, Anti-Caspase1, TNF-α were purchased from R&D Systems (Minneapolis, MN, USA); Anti-cleaved caspase1, Anti-NLRP3, Anti-IκBα, Anti-phospho-P38, Anti-phospho-ERK, Anti-phospho-JNK, Anti-phospho-IKKα/β, and Anti-Tublin were purchased from Cell Signaling Technology (Boston, MA, USA); Anti-TAK1, Anti-phospho-TAK1, Anti-ASC and recombinant active caspase1 protein were purchased from Abcam (Shanghai, China). The ultrapure LPS and Nigericin were obtained from Sigma-Aldrich (St. Louis, MO, USA). The caspase1 activity assay kit was purchased from Beyotime Biotechnology (Shanghai, China). The Human IL-1β ELISA Kit was purchased from Boster Biological Technology (Wuhan, China).

#### 3.3.2. Cell preparation and Stimulation

Human THP-1 and HEK-293 from the American Type Culture Collection (Manassas, VA, USA) were grown in RPMI 1640 medium and DMEM respectively, supplemented with 10% inactivated fetal bovine serum and 1% penicillin-streptomycin (Invitrogen, Waltham, MA, USA). 293 cells were digested with 0.05% trypsin-EDTA and split twice a week. 

3 × 10^5^ THP-1 cells were plated in 24-well plates or 1.5 × 10^6^ THP-1 plated in 6-well plates and primed with 50 nM PMA for 24 h and incubated with or without tested derivatives overnight. The cells were primed for 3 h with ultrapure LPS (5 µg/mL), and then stimulated for 45 min with nigericin (20 µM) in serum-free medium. Cell extracts and precipitated supernatants were analyzed by immunoblot or ELISA. 293 cells were incubated with or without target derivatives for 2 h followed by stimulated with TNF-α (20 ng/mL).

#### 3.3.3. Western Blot

Cells were washed with PBS and were lysed in lysis buffer (20 mM Tris-HCl, pH 7.5, 150 mM NaCl, 10 mM β-glycerophosphate, 5 mM EGTA, 1 mM sodium pyrophosphate, 5 mM NaF, 1 mM Na3VO4, 0.5% Triton X-100, and 1 mM DTT) supplemented with protease inhibitors (1 mM PMSF, 5 μg/mL leupeptin, 5 μg/mL pepstatin A, and 5 μg/mL aprotinin). The supernatants were treated with 1/4 volume TCA to precipitate protein, followed by centrifuge and washed with acetone. Proteins were separated by SDS-PAGE and were electrically transferred to a polyvinylidene difluoride membrane. The membranes were blocked for one hour at room temperature in TBS-T (150 mM NaCl, 20 Mm Tris, pH 7.5, 0.05% Tween-20) containing 5% milk and probed with specific first antibodies for one hour at room temperature. Blots were then incubated with HRP-conjugated anti-rabbit (7074, Cell Signaling) or anti-mouse (7076, Cell Signaling Technology) secondary antibody. Immunoreactivity proteins were visualized using the ECL detection system (Bio-Rad, Hercules, CA, USA).

#### 3.3.4. Cytotoxicity Assay

THP-1 cells were seeded into 96-well plates at 1.5 × 10^4^ cells per well with PMA (50nM). After 24 h incubation, fresh culture medium containing test compounds at various concentrations were added. 24 h later, cytotoxicity was evaluated with 3-(4,5-Dimethylthiazol-2-yl)-2,5- diphenyltetrazolium bromide (MTT).

#### 3.3.5. Immunofluorescence

THP-1 were plated in 24-well plates and stimulated as previously described. Following the treatment, cells were fixed by 4% paraformaldehyde and permeabilized by 0.3% Triton X-100. After blocking with 1% BSA for 1 h, cells were stained with ASC antibodies overnight at 4 °C. After three washes with PBS, cells were stained with Alexa Fluor 484 secondary antibodies (Abcam) for another hour. 4′,6-Diamidino-2-phenylindole (DAPI) were used to stain nuclei. Images were recorded on a fluorescence microscope.

#### 3.3.6. Statistics

Results are presented as mean values ± standard error of independent triplicate experiments. All statistical analyses were performed by using two-tailed Student’s t-test and *p*-values of less than 0.05 were considered statistically significant.

### 3.4. Molecular Modeling Analysis

Discovery Studio 4.5 software was used for the preparation of the ligand and receptor. An automated docking study was carried out using the 3D structure of NLRP3^pyrin^ (PDB code: 3Q2F), ASC^Pyrin^ (PDB code: 3J63) and ASC^CARD^ (PDB code: 6N1H). The regularized protein was used in determination of the important amino acids in the predicted binding pocket. Interactive docking using CDOCKER protocol was carried out for all the conformers of ligand **6** to the selected active site, after energy minimization using prepare ligand protocol. The docked compound was assigned a score according to its binding mode onto the binding site.

## 4. Conclusions

To conclude, a series of 7-substitued fangchinoline analogues, of which twenty two were novel, were synthesized and evaluated for their effect to inhibit LPS/NIG-induced IL-1β release at both cell and protein levels at the concentration of 5 μM. Among them, compound **6** exhibited promising potency against IL-β activation with an inhibition rate of 76.1% at a concentration of 5 μM and an IC_50_ value of 3.7 μM. A preliminary mechanism study revealed that **6** failed to block NF-κB and MAPK pathways, and was not a direct caspase-1 inhibitor either. It might combine with NLRP3, and inhibit its ability to recruit ASC and finally inhibit IL-1β release, as described in Figure 7. Our current study supports the potential role of compound **6** against inflammatory diseases, and these fangchinoline derivatives offer powerful information for further strategic optimization and provide promising potency for the development of a novel class of anti-inflammatory agents.

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
