# Peer review of "Synthesis and Biological Evaluation of Fangchinoline Derivatives as Anti-Inflammatory Agents through Inactivation of Inflammasome"

_molecules, 2019, doi:10.3390/molecules24061154_

Round 1

Reviewer 1 Report

The study is immersed and beautifully presented, the results being encouraging for the future of anti-inflammatory therapy. Please keep in your attention the following suggestions:

The Abstract is much too similar to the Conclusions. I suggest reformulation.

It should be noted that how many fangchinoline derivatives were synthesized to highlight the degree of novelty by synthesis

In summary, it is preferable to mention fangchinoline derivatives instead of fangchinoline compounds

The compound number must be written bold (6)

Page 1 Type must be written type

Page 2 Figure 1 Please modify Figure 1 so that the molecular modeling strategy is more obvious with the preservation of atom positions

Page 2  I suggest the replacement od deacid agent with acid neutralizer

Page 2 N-(3-bromopropyl) phthalimide must be written N-(3-bromopropyl)phthalimide

Page 3 Scheme 1 must be modified °C (to read better)

Page 3 The chemical structures in the table 1 should be written larger (for compounds 3e, 3f……)

Please write in the entire paper mL

Page 4 amino propyl must be written aminopropyl

Page 5 As shown must be written as shown

Page 7 van der waals must be written Van der Waals

Page 8 “After removal of the solvent in vacuo, the residue was purified over by flash” just in vacuo must be written italic

Page 10 C48H50N3O8 must be written as formula

Page 15 hoechst must be written Hoechst

Page 16 I suggest to use fangchinoline derivatives instead of fangchinoline compounds. Please reformulate to avoid being more like what you have written in the abstract

Author Response

 Q1: The Abstract is much too similar to the Conclusions. I suggest reformulation.

R: Thanks for the suggestion, and we have reformulated the conclusion to avoid this problem.

Q2: It should be noted that how many fangchinoline derivatives were synthesized to highlight the degree of novelty by synthesis

R: Thanks for the suggestion, and we have noted the number in the revised version.

Q3: In summary, it is preferable to mention fangchinoline derivatives instead of fangchinoline compounds.

R: Thanks for the advice, and we have replaced fangchinoline compounds with fangchinoline derivatives.

Q4: The compound number must be written bold (6)

R: We are sorry for the mistake, and we have revised it.

Q5: Page 1 Type must be written type.

R: We are sorry for the mistake, and we have revised it.

Q6: Page 2 Figure 1 Please modify Figure 1 so that the molecular modeling strategy is more obvious with the preservation of atom positions

R: Thanks for the suggestion, yes, figure 1 was modified as instructed.

Q7: Page 2  I suggest the replacement od deacid agent with acid neutralizer

R: Thanks for the suggestion, and we have replaced it as instructed.

Q8: Page 2 N-(3-bromopropyl) phthalimide must be written N-(3-bromopropyl)phthalimide

R: Thanks for the suggestion, and we have revised it.

Q9: Page 3 Scheme 1 must be modified °C (to read better)

R: Thanks for the suggestion, we have modified scheme 1 and rewrote °C.

Q10:Page 3 The chemical structures in the table 1 should be written larger (for compounds 3e, 3f……)

R: Thanks for the suggestion, and we have modified them into larger sizes.

Q11: Please write in the entire paper mL

R: We are sorry for the mistake, and we have corrected them.

Q12: Page 4 amino propyl must be written aminopropyl

R: Thanks for the correction, and we have revised it.

Q13: Page 5 As shown must be written as shown

R: Thanks for the correction, and we have revised it.

Q14: Page 7 van der waals must be written Van der Waals

R: Thanks for the correction, and we have revised it.

Q15: Page 8 “After removal of the solvent in vacuo, the residue was purified over by flash” just in vacuo must be written italic

R: Thanks for the correction, and we have revised it.

Q16: Page 10 C48H50N3O8 must be written as formula

R: Thanks for the correction, and we have revised it.

Q17: Page 15 hoechst must be written Hoechst

R: Thanks for the correction, and we realized the problem. In fact, another dye instead of hoechst was used, and we have corrected it as “4',6-Diamidino-2-phenylindole (DAPI)”.

Q18: Page 16 I suggest to use fangchinoline derivatives instead of fangchinoline compounds. Please reformulate to avoid being more like what you have written in the abstract

R: Thanks for the suggestion, we have replaced it, and reformulate the conclusion.

Reviewer 2 Report

The experimental design does not include statistical analysis of the results. Thus, there is a lack of scientific rigour in  this manuscript. 

Selection of compounds is based on cytotoxicity but wrong results without statistics are presented. The authors tested some concentrations of compounds which have not been tested for cytotoxicity. 

The experimental design of this work is extremely wrong. 

Author Response

Q: The experimental design does not include statistical analysis of the results. Thus, there is a lack of scientific rigour in this manuscript. Selection of compounds is based on cytotoxicity but wrong results without statistics are presented. The authors tested some concentrations of compounds which have not been tested for cytotoxicity. The experimental design of this work is extremely wrong. 

R: We thank the reviewer for this constructive comment. According to the reviewer’s suggestion, cytotoxic data and IL-1β inhibition rate data were performed statistical analyses by using two-tailed Student's t-test. Besides, all compounds exhibiting the inhibitory rates of over 60% (instead of 50%) in Table 1 were gathered to evaluate their cytotoxic effects in THP-1 cells, and the cell proliferation results in original Fig.3A was revised accordingly. We are grateful for the reviewer to point out this mistake. In addition, in order to make the process of compounds selection more clearly, we exchanged Fig.2B with Fig.3A, and the figure legends and results section have been revised accordingly. It will be greatly appreciated if the reviewer and the editor could understand our idea.

Reviewer 3 Report

The manuscript by Li , Deng and collaborators: Synthesis and Biological Evaluation of Fangchinoline Derivatives as Anti-inflammatory Agents through Inactivation of Inflammasome

is an important contribution in the field of inflammation.

The manuscript requires Extensive editing of English language.

What means "sodium hydride as the deacid agent"? page 2

Yields should be also listed in mg and not only in %.

Bruker Avance 400 spectrometer or Bruker Avance III 500 spectrometer (Varian, San

Francisco, USA) should be corrected not Varian. page 7.

AutospecUitima-TOF spectrometer should be Autospec Ultima page 7.

Author Response

Q1:The manuscript requires Extensive editing of English language.

R: Thanks for the comment. We have revised the manuscript thoroughly and corrected several mistakes.

 Q2: What means "sodium hydride as the deacid agent"? page 2

R: Thanks for the question, and we have replaced “deacid agent” with “acid neutralizer”.

Q3: Yields should be also listed in mg and not only in %.

R: Thanks for the reminder, and we have added the weight information in the revised version.

Q4: Bruker Avance 400 spectrometer or Bruker Avance III 500 spectrometer (Varian, San

Francisco, USA) should be corrected not Varian. page 7.

R: We are sorry for the mistake, and we have corrected it in the revised version.

Q5: AutospecUitima-TOF spectrometer should be Autospec Ultima page 7.

R: Thanks for the correction, and we have revised it.

Round 2

Reviewer 3 Report

The authors addressed all the points from the reviewer. The manuscript was changed accordingly.

I recommend the acceptance of the manuscript to be published in the journal.